# Genome-Wide Identification and Expression Analysis of the Starch Synthase Gene Family in Sweet Potato and Two of Its Closely Related Species

**DOI:** 10.3390/genes15040400

**Published:** 2024-03-25

**Authors:** Zongjian Sun, Zhenqin Li, Xiongjian Lin, Zhifang Hu, Mengzhen Jiang, Binquan Tang, Zhipeng Zhao, Meng Xing, Xiaohui Yang, Hongbo Zhu

**Affiliations:** College of Coastal Agricultural Sciences, Guangdong Ocean University, Zhanjiang 524088, China; szj206131@outlook.com (Z.S.); lizhenqiny@126.com (Z.L.); l1843204672@outlook.com (X.L.); 2112204001@stu.gdou.edu.cn (Z.H.); funkhly@ofice.vip (M.J.); tang441823@outlook.com (B.T.); beishu@foxmail.com (Z.Z.); xm18729371511@163.com (M.X.); yangxin199511@163.com (X.Y.)

**Keywords:** sweet potato, *Ipomoea trifida*, *Ipomoea triloba*, starch synthase, abiotic stress, bioinformatic analysis

## Abstract

The starch synthase (SS) plays important roles in regulating plant growth and development and responding to adversity stresses. Although the *SS* family has been studied in many crops, it has not been fully identified in sweet potato and its two related species. In the present study, eight *SSs* were identified from *Ipomoea batatas* (*I. batata*), *Ipomoea trifida* (*I. trifida*), and *Ipomoea trlioba* (*I. trlioba*), respectively. According to the phylogenetic relationships, they were divided into five subgroups. The protein properties, chromosomal location, phylogenetic relationships, gene structure, cis-elements in the promoter, and interaction network of these proteins were also analyzed; stress expression patterns were systematically analyzed; and real-time polymerase chain reaction (qRT-PCR) analysis was performed. *Ipomoea batatas* starch synthase (*IbSSs*) were highly expressed in tuber roots, especially *Ipomoea batatas* starch synthase 1 (*IbSS1*) and *Ipomoea batatas* starch synthase 6 (*IbSS6*), which may play an important role in root development and starch biosynthesis. At the same time, the *SS* genes respond to potassium deficiency, hormones, cold, heat, salt, and drought stress. This study offers fresh perspectives for enhancing knowledge about the roles of *SSs* and potential genes to enhance productivity, starch levels, and resistance to environmental stresses in sweet potatoes.

## 1. Introduction

Starch is the second-most abundant carbohydrate on earth and an important energy source for human and animal nutrition. Starch mainly consists of amylose (AM) and branched-chain amylopectin (AP) [1,2,3]. AM/AP affects the pasting temperature, solubility, and palate of starch, while the composition of starch is mainly affected by the enzyme starch synthase [4,5,6]. Starch synthase (SS) contains soluble starch synthases (*SSS*) and granule-bound starch synthase (GBSS), which are highly similar in the C-terminal region and are key players in starch synthesis [7,8,9]. The SSSs are usually composed of a core region with a molecular weight of about 60 kDa, which is essential for catalytic activity, and consist of two conserved structural domains, including the Glyco_transf_5 structural domain (GT5, PF08323) and the Glycos_transf_1 structural domain (GT1, PF00534) [7,10,11,12,13]. The SSs contained Glyco_transf_1 (GT1; PF00534) and Glyco_transf_5 (GT5; PF08323) structural domains [7,11,13,14].

Starch synthesis is a complex biological process, requires the synergistic involvement of several enzymes, and is subject to environmental and developmental regulation [7]. *GBSS* is a key gene in AM synthesis, with *GBSSI* functioning mainly in endosperm and pollen tissues and *GBSSII* functioning mainly in leaves and photosynthetic tissues. At the same time, *GBSS* is also involved in AP synthesis. For example, by suppressing the expression of *GBSSI*, the content of AM was reduced, the long chains above degree of polymerization (DP) 100 were lacking in AP, and the chains of DP 6 and DP 7 were slightly reduced in AP; by up-regulating the expression of *GBSSI*, the content of AM was increased [1]. These were validated in potato, sweet potato, wheat, and rice [15,16]. *SSSs* play an important role in the process of AP synthesis, in which *SSSI* is mainly responsible for prolonging the short-branched side chains of DP 6 or DP 7 to DP 8–12 and is also involved in the synthesis of the external short chains of starch clusters [10,17]. The elongated branches of *SSSI* can be further extended to intermediate lengths by *SSSII* and alter the content of branched starch in the crop [17]; *SSSIII* produces longer chains that extend between short-chain polysaccharide clusters [8]; *SSSIV* was expressed predominantly in leaves, has low sequence identity to starch synthases expressed in the endosperm, and was involved in the process of starch granule initiation or at least controls the number of starch granules in the chloroplasts [18,19,20,21]; *SSSV* was expressed mainly in spike leaves and seeds, synthesizing transient and storage starch, respectively, and its transcript reaches its maximum at the seed filling stage, which was important for seed starch synthesis [22]. The ratio of AM/AP has an important role in crop quality, flavor, and quality, as well as stress tolerance, and is important for promoting crop yield and stress tolerance. Presently, the *SS* gene family members have been studied in several plants, such as *Arabidopsis* [23], rice [24], potato [25,26], and tomato [27].

Sweet potato (*Ipomoea batatas* (L.)), the world’s seventh largest food crop, is a dicotyledonous annual or perennial plant. It belongs to the family Convolvulaceae, which is grown on a large scale around the world because of its high yield and resistance to abiotic stresses. It is an important root crop, and the main component of its storage roots is starch, which accounts for 50–80% of the dry weight [6] and provides humans with abundant carbohydrates, dietary fiber, vitamins, and trace elements [5,28]. Some studies have shown that inhibition of *IbGBSSI* gene expression by RNA interference dramatically suppresses AM expression [15,16]. Overexpression of *IbSSI* affects sweet potato starch content, grain size, and proportion of branched-chain starch [29]. Inhibition of *IbSSII* gene expression by RNA interference dramatically alters the basic structure of starch and thus its functional properties [30]. Furthermore, *IbGBSSI* plays an important role in different drought-tolerant cultivars under drought stress [31]. Therefore, *SSs* may be involved in starch synthesis and abiotic stress responses in sweet potato. However, the SS gene family members have not been identified in sweet potato. Meanwhile, with the release of the genome assemblies of sweet potato and two diploid wild relatives, *I. trifida* and *I. triloba*, it will be possible to identify and analyze the *SS* gene family of sweet potato [28].

A total of 24 *SS* genes were successfully pinpointed in sweet potato as well as their two diploid wild counterparts. The results of the collinearity analysis showed that there was a strong correlation between sweet potato and its two relatives. The analysis of real-time fluorescence PCR and expression profiles revealed the significant involvement of *SS* genes in plant growth and response to various environmental challenges such as salt stress and heat stress. In conclusion, our results lay the groundwork for future exploration into the functionalities of sweet potato *SS* genes.

## 2. Results

### 2.1. Identification of SS in Sweet Potato and Two Closely Related Species

Eight *SS* genes were identified in *I. batatas*, *I. trifida*, and *I. triloba*, respectively, named after “*IbSS1-8*”, “*ItfSS1-8*”, and “*ItbSS1-8*”, respectively, according to their positions on the chromosomes (Table 1). Physicochemical properties analysis showed that the amino acid number of IbSSs ranged from 633 to 1402, the number of amino acids of ItfSSs ranged from 538 to 1391, and the number of amino acids of ItbSSs ranged from 608 to 1349. The molecular weights of *IbSSs* ranged from 70.69 to 157.43 kDa; the molecular weights of ItfSSs and ItbSSs ranged from 59.38 to 156.75 kDa and 66.70 to 152.42 kDa, respectively. The theoretical potentials of SSs in *I. batatas* ranged from 5.11 to 7.15, the theoretical potentials of ItfSSs and ItbSSs ranged from 4.96 to 6.56 and 5.13 to 8.31, respectively. In sweet potato, all IbSS genes were unstable except for *IbSS2*, *IbSS6,* and *IbSS7*. *ItfSS2*, *ItfSS4*, *ItfSS5,* and *ItfSS8* were stable; the other ItfSS genes were unstable; and *ItbSS2*, *ItbSS5,* and *ItbSS8* were stable ones. The hydrophilicity of IbSSs ranged from −0.561 to −0.023, the hydrophilicity of ItfSSs ranged from −0.571 to 0.035, and the hydrophilicity of *ItbSSs* ranged from −0.579 to 0.075. Overall, there was significant variability in the physicochemical properties of *I. batatas*, *I. trifida*, and *I. triloba.*

The chromosomal localization analysis showed that *SSs* in *I. batatas*, *I. trifida*, and *I. triloba* were separately distributed on 5 chromosomes (Figure 1A–C). In sweet potato, two *SS* genes were detected in LG5, LG7, and LG11, and one *SS* gene was detected in LG9 and LG10; *I. trifida* and *I. triloba* showed similar distributions in Chr01, Chr03, Chr08, Chr10, and Chr12, respectively, and it was hypothesized that there might be a one-to-one correspondence. In addition, a collinearity analysis of each species between *I. batatas*, *I. trifida*, *I. triloba*, *Arabidopsis thaliana (A. thaliana*), *Solanum tuberosum* L. (*S. tuberosum)*, and *Oryza sativaL (O. sativa*) was generated. As shown in Figure 1D, a strong collinearity relationship occurred among *I. batatas*, *I. trifida*, and *I. triloba*, and a weak collinearity relationship occurred between *I. batatas* and other varieties. Among these genes, *IbSS7* and *IbSS8* were homologous to *ItfSS1*, *ItSS2*, *ItbSS1,* and *ItbSS2*, respectively; *IbSS6* was homologous to *ItfSS5* and *ItbSS5*; *IbSS1* and *IbSS2* were related to *ItfSS7*, *ItfSS8*, *ItfSS3*, *ItfSS4*, and *ItbSS7*, *ItbSS8*, *ItbSS3*, and *ItbSS4*, respectively; *IbSS3* and *IbSS4* were homologous to *ItfSS7*, *ItfSS8*, *ItfSS3*, *ItfSS4*, and *ItbSS7*, *ItbSS8*, *ItbSS3*, and *ItbSS4*, respectively; and *IbSS5* was homologous to *ItfSS6* and *ItbSS6.* These results suggested that segmental duplication was present in the evolutionary process from diploid to hexaploid.

### 2.2. Analysis of Phylogenetic Relationships between Sweet Potato and Two Diploid Wild Relatives

In order to study the phylogenetic relationship of *SSs* among *I. batatas*, *I. trifida*, *I. triloba*, *A. thaliana*, *S. bicolor*, *O. sativa*, and *S. lycopersicum*, a phylogenetic tree was constructed using MEGA-X, and 55 *SSs* were classified into five groups (Figure 2). *SSs* were distributed in varying quantities in all groups; the specific distributions were as follows: (total: *I. batatas*, *I. trifida*, *I. triloba*, *A. thaliana*, *S. bicolor*, *O. sativa*, and *S. lycopersicum*): SSI (7: 1, 1, 1, 1, 1, 1, and 1); SSII: (14: 2, 2, 2, 1, 3, 3, and 1); SSIII (9: 1, 1, 1, 1, 2, 2, and 1); SSIV (17: 3, 3, 3, 1, 3, 3, and 1); GBSS (8: 1, 1, 1, 1, 2, 1, and 1). Additionally, each *IbSS* had a homologous gene in *I. trifida* and *I. triloba*, respectively.

### 2.3. Conserved Motifs and Exon–Intron Structure Analysis of SS in Sweet Potato and Two Closely Related Species

The protein sequences of *I. batatas*, *I. trifida*, and *I. triloba* were analyzed for conserved motifs by the MEME online tool and Tbtools software, and 10 conserved motifs were identified (Figure 3). The analysis showed that most of the *SSs* contained these 10 conserved motifs; however, it varies in different subgroups; in group II, *IbSS1* and *IbSS4* lacked 1 and 2 motifs, respectively. In group III, *IbSS8*, *ItfSS1*, and *ItbSS1* lacked 2, 1, and 1 motifs, respectively. In group IV, *IbSS5* and *ItbSS4* lack 2 and 3 motifs, respectively. *ItbSS4*, which contained 7 conserved motifs; *IbSS4*, *IbSS5,* and *IbSS8*, which contained 8 conserved motifs; *ItfSS1*, *ItbSS1,* and *IbSS1*, which contained 9 conserved motifs.

Analyzing the exon–intron distribution of *IbSS*, *ItfSS*, and *ItbSS* provides a better understanding of the evolution of the *IbSS* gene family. Overall, the number of exons and introns in *SSs* was highly variable (Figure 3). The number of exons ranged from 9 to 24. Additionally, the *SSs* in the same group have the same number of exons, such as *ItfSS3*, *ItbSS3*, *IbSS1*, and *ItbSS7* in the SSII group (all containing 10 exons) and *IbSS8*, *ItfSS1*, *ItbSS1*, *ItbSS1*, *ItfSS6*, and *ItbSS6* in the SSIII and SSI groups (all containing 18 exons). Interestingly, some exons were more abundant in sweet potato: *IbSS5* in IV has 19 exons, while both *ItfSS6* and *ItbSS6* contain 18 exons; *IbSS3* contains 24 exons, while *ItfSS4* and *ItbSS4* contain 18 and 22 exons, respectively; *IbSS5* contains 19 exons, while both *ItfSS6* and *ItbSS6* contain 18 exons. These results suggested that the differences between exons and introns may lead to evolutionary and functional differences among *SS*s, and the *SS* gene family may have become more complex during evolution.

### 2.4. Analysis of Cis-Acting Elements of SS Promoters in Sweet Potato and Two Closely Related Species

Plant promoter cis-acting elements were closely related to plant development, hormone regulation, and response to adversity. A large number of core promoter elements and generic homeopathic elements were distributed in the promoter region of *I. batatas*, *I. trifida*, and *I. triloba*. Based on the functions of the components, we classified them as core promoter elements, light-responsive elements, hormone responsive elements, and stress response elements, respectively (Figure 4 and Appendix A). *SSs* have a variety of hormonal response elements, such as abscisic acid response element ABRE; auxin response elements TGA and AUxR; and gibberellin response elements P-box, GARE, and TATC-box. In addition, there were abiotic stress response elements, for example, the low-temperature response element LTR; elements that were essential for anaerobic induction; drought response elements; and salt response elements MBS, MYB, and MYC. This indicates that plant growth and development, hormone action, light regulation, and abiotic/biotic stress responses in sweet potato and its two relatives were controlled by *SSs*.

### 2.5. Expression Analysis of SS Genes in Sweet Potato and Two Closely Related Species

#### 2.5.1. Expression Analysis in Different Tissues

Based on the transcriptome data from sweet potato flower, fruit, leaf, stem, primary root, firewood root, fibrous root, and tuber root, a sweet potato *SS* gene expression heatmap was generated (Figure 5A). We found that these *IbSSs* had different levels of expression in the eight tissues of *I. batatas*. As can be seen from Figure 5A, seven *IbSSs* were highly expressed in tuber root, such as *IbSS6*, *IbSS1*, and *IbSS8*. The SSs were more expressed in stems and leaves, such as *IbSS1*, *IbSS5*, and *IbSS6.* Overall, the *IbSS6* gene was highest expressed in all tissues of sweet potato except in leaves, where it was lower than the *IbSS5* gene.

We analyzed the expression of *SSs* in *I. trifida* and *I. triloba* (Figure 5B,C). In *I. trifida*, six genes were highly expressed in root1, especially *ItfSS5* and *ItfSS7*; *ItfSSs* were also highly expressed in leaves, such as *ItfSS5* and *ItfSS8*. Similarly, in *I. triloba*, four genes were highly expressed in flower buds, such as *ItbSS5* and *ItbSS7*; *ItbSSs* were also expressed at a higher level in leaves and root1. Overall, *SS5* and *SS7* were highly expressed in *I. trifida* and *I. triloba*.

#### 2.5.2. Expression Analysis under Potassium Deficiency in Sweet Potato

Figure 6 displays the expression patterns of the eight *SS* genes in sweet potato, which were identified through transcriptome analysis of the low-k-tolerant “Xushu 32” and low-k-sensitive “Ningzishu 1” varieties with contrasting. The data were retrieved from the PRJNA1013090 dataset in the NCBI repository. During a low-potassium treatment, with the exception of *IbSS4*, which showed decreased expression in the two cultivars, the remaining seven *SSs* exhibited increased expression in both varieties.

#### 2.5.3. Expression Analysis under Hormone Stress

Using the “Xushu 18” RNA-seq data retrieved from the NCBI database (PRJNA511028), the expression patterns of eight *SS* genes in sweet potato were analyzed in three different tissues following treatments with ABA, SA, and MeJA (Figure 7). In fibrous roots, the expression of eight *SS* genes was up-regulated after ABA treatment; among them, the expression level of the *IbSS6* gene was particularly high; *IbSS1* and *IbSS6* genes were down-regulated after SA treatment; other genes showed no changes; *IbSS1*, *IbSS4,* and *IbSS6* were up-regulated after MeJa treatment; among them, *IbSS6* was significantly increased in the fibrous roots. In stems, *IbSS3* and *IbSS8* decreased after ABA treatment; there was no change before and after *IbSS5*, the other genes were up-regulated to varying degrees, among which *IbSS1* and *IbSS6* were significantly up-regulated; under SA treatment, except for *IbSS3*, all other genes were up-regulated, particularly *IbSS1*, *IbSS2*, *IbSS6,* and *IbSS8* were significantly increased; after MeJa treatment, all *IbSSs* genes were up-regulated; among them, *IbSS1*, *IbSS6* and *IbSS7* have been significantly increased. In leaves, under ABA and SA treatment, *IbSS6* was downregulated, and other genes were upregulated; after MeJa treatment, *IbSS2*, *IbSS4*, *IbSS6*, and *IbSS8* were downregulated, and *IbSS1* and *IbSS7* were raised.

We also utilized the expression levels of *I. trifida* and *I. triloba* under ABA, GA, and IAA treatments. After ABA treatment, *ItfSS7*, *ItbSS3*, and *ItbSS7* were up-regulated. After GA treatment, *ItfSS1*, *ItfSS8*, *ItbSS2*, *ItbSS4*, and *ItbSS6* were up-regulated. Under IAA treatment, *ItfSS5*, *ItbSS1*, *ItbSS3*, *ItbSS4*, *ItbSS6*, and *ItbSS8* were up-regulated (Figure 8).

#### 2.5.4. Expression Analysis under Cold Stress

Figure 9 presents the detected expression profiles of the eight *IbSSs*, using transcriptome data from “Shenshu 28” (a cold-sensitive variety) and “Liaohanshu 21” (a cold-tolerant variety) under cold stress conditions. Under cold stress, in “Shenshu 28”, *IbSS4* and *IbSS6* were up-regulated, and in “Liaohanshu 21”, *IbSS2*, *IbSS4*, *IbSS6*, and *IbSS8* were up-regulated.

Furthermore, the expression patterns of *ItfSSs* and *ItbSSs* were examined in response to cold stress (see Figure 10). With the exception of *ItfSS5* and *ItbSS2*, which showed an increase in expression, the remaining genes demonstrated a decrease in expression levels when subjected to cold stress.

#### 2.5.5. Expression Analysis under Heat Stress

We used the heat-tolerant variety “Guangshu 87” and the heat-sensitive material “Ziluolan” to detect the expression profiles of *SS* genes in sweet potatoes after heat stress (Figure 11). After heat treatment, the expression levels of the *IbSS3*, *IbSS4*, and *IbSS5* genes in the fibrous roots of the “Ziluolan” cultivar were up-regulated, while the other genes were down-regulated. Except for *IbSS3* and *IbSS5*, all other genes were down-regulated in the tuberous roots. *IbSS3* and *IbSS5* were up-regulated in the fibrous roots of the “Gunagshu 87” cultivar, and all other genes were down-regulated, except for *IbSS3*. The other genes were down-regulated in the tuber roots.

We also analyzed the expression levels of *ItfSSs* and *ItbSSs* under 35 °C heat stress (Figure 12). After heat treatment, except for *ItfSS3*, *ItfSS4*, *ItfSS7*, *ItbSS3*, *ItbSS4*, and *ItbSS7*, the transcript levels of the others were increased.

#### 2.5.6. Expression Analysis under Salt and Drought Stresses

As shown in Figure 13, we analyzed the expression levels of *IbSSs* in the leaf, primary root, and stem under salt and drought treatments. In leaves, after salt treatment, *IbSS1* and *IbSS6* were up-regulated, and other *IbSSs* were down-regulated; under drought treatment, *IbSS1* and *IbSS2* were up-regulated, while other *IbSSs* were down-regulated. In the primary root, after salt treatment, *IbSS4* and *IbSS6* were up-regulated, and other genes were down-regulated; after drought treatment, *IbSS4* was adjusted upward, while other *IbSSs* were downgraded. In stems, except for *IbSS5*, which was up-regulated in the drought treatment, the other genes were up-regulated in both salt and drought treatments.

We analyzed the expression levels of *ItfSSs* and *ItbSSs* under drought and salt stress to understand the role of *ItfSSs* and *ItbSSs* in two closely related species (Figure 14). In *I. trifida* (Figure 14A), after salt treatment, *ItfSS1* and *ItfSS5* were down-regulated, and other *ItfSSs* were up-regulated; under drought treatment, *ItfSS1*, *ItfSS5*, and *ItfSS6* were down, and other *ItfSSs* were raised. In *I. triloba* (Figure 14B), under salt treatment, *ItbSS2*, *ItbSS6*, and *ItbSS7* were regulated upward, *ItbSS4* remained unchanged, and other *ItbSSs* were down-regulated; after drought treatment, *ItbSS2*, *ItbSS4*, *ItbSS7*, and *ItbSS8* were adjusted upward, while other *ItbSSs* were downgraded.

### 2.6. Real-Time PCR Analysis of SS Genes in Sweet Potato

Two representative varieties of “J26” with high amylose content and “P32” with low amylose content were selected, and qRT-PCR was carried out to detect and analyze the expression of *SSs* in the stem, leaf, flower, seed, and root at four different stages (Figure 15). In root1, except that *IbSS2* and *IbSS6* were expressed higher in “J26” than in “P32”, the expression levels of other *IbSSs* in “P32” were higher than those in “J26”. In root2, except that *IbSS6* was expressed higher in “J26” than in “P32”, the expression levels of other *IbSSs* in “P32” were higher than those in “J26”. In firewood root and flower, except that *IbSS1* was expressed higher in “J26” than in “P32”, the expression of other *IbSSs* was approximately the same in both varieties. In fibrous roots, except that the expression of *IbSS6* in both varieties was about the same, the expression levels of other *IbSSs* in “P32” were higher than those in “J26”. In stems, the expression of *IbSSs* in “P32” was higher than that in “J26”. In leaves, except that *IbSS2*, *IbSS4*, and *IbSS5* were expressed higher in “J26” than in “P32”, the expression levels of other *IbSSs* in “J26” were lower than those in “P32”. In seed, except that *IbSS2* and *IbSS6* were expressed higher in “Jishu 26” than in “P32”, the expression levels of other *IbSSs* in “J26” were lower than those in “P32”. These results suggested that *IbSSs* may play an important role in different stages of root development. At the same time, *SSs* had a certain effect on the growth and development of stems and leaves, which may further affect the yield and quality of sweet potato by affecting photosynthesis.

### 2.7. Sweet Potato IbSSs Protein Interactions

To investigate the potential regulatory network of SS, we constructed an IbSS interaction network map based on *Arabidopsis* homologous proteins (Figure 16). IbSS could interact with glucose-1-phosphate adenylyltransferases (ADG2, APS1, APL2, APL3, and APL4), dismutases (DPE1, DPE2), starch branching enzymes (SBE2.1, SBE2.2, SBE3), and UDP-glycosyltransferases (A0A1P8B9B9). Protein interaction predictions indicated that all eight IbSSs were homologous to *Arabidopsis* proteins, IbSS2 was homologous to SS1; IbSS1 and IbSS4 were homologous to SS2; Ib*SS3*, IbSS5, and IbSS7 were homologous to SS4; IbSS6 was homologous to GBSS1; and the homology with SS4 was also up to 31.4%. These results suggest that *IbSS* plays an important role in the metabolic development of sweet potato.

## 3. Discussion

### 3.1. Evolution of SS in Sweet Potato and Two Closely Related Species

Starch is widely found in green plants and is an important carbon and energy reserve in plants, which is essential for the yield and quality of sweet potato lakes. The *SS* gene family has been identified in many plants, such as *Arabidopsis* containing five *SS* genes [13,23], rice containing 10 *SS* genes [24], potato containing four *SS* genes [11], maize [11], and wheat [32,33], and tomato [27] containing six *SS* genes. In this study, 24 *SS* genes from *I. batatas*, *I. trifida*, and *I. triloba* were grouped into five subfamilies based on a phylogenetic tree. These genes had strong collinearity between *I. batatas*, *I. trifida*, and *I. triloba*; the homologs between the three species were located at similar points (Figure 1D and Figure 2). This suggests that sweet potato *SS* originated from its diploid relatives. As Figure 3 showed, except for *IbSS3* and *IbSS5*, 10 conserved motifs were identified for all other *IbSSs*. *IbSS3*, *ItfSS4*, and *ItbSS4* are homologous, but *IbSS3* does not have motif 4 or motif 2. *IbSS5*, *ItfSS6*, and *ItbSS6* are homologous, but *IbSS5* does not have motif 2, motif 6, or motif 9. This may be since there may have been some changes in the function of the sweet potato *SS* genes during evolution.

The structures of gene exons and introns are generally preserved in homologous genes within a gene family [34]. Our research found that most homologous sequences in *I. batatas*, *I. trifida*, and *I. triloba* shared similar exon and intron counts. However, there was some difference; for example, in the SSIV group, *IbSS3* contains 24 exons, *ItfSS4* and *ItbSS4* contain 18 and 22 exons, respectively; *IbSS7* contains 19 exons, and *ItfSS2* and *ItbSS2* contain 18 exons. Other than that, in the SSI group, *IbSS2* has two fewer exons than *ItfSS8* and *ItbSS8*; in the SSII group, *IbSS4* has two fewer exons than *ItfSS3* and *ItbSS3*; and in the GBSS group, *IbSS6* has one fewer exon than *ItbSS5*. This suggests that there may have been some changes in the function of *SSs* during the evolution of sweet potato and its relative.

### 3.2. Distinct Roles of SS Genes in Biological Processes

The distribution of *SSs* in various tissues, stages of development, and exposure to environmental stresses provided valuable insights into the elucidation of their inherent biological roles [35]. Overall, the expression of *IbSSs* was higher in the tuberous roots of two cultivars with different amylose contents, and the expression levels of *IbSS1* and *IbSS6* were the highest in the tuberous roots. This suggests that the *IbSS1* and *IbSS6* genes mainly acted in tuberous root development and starch biosynthesis. The *IbSS1* and *IbSS6* expression levels were also higher in the stems and leaves (Figure 5A). This suggested that the *IbSS1* and *IbSS6* genes may participate in transport, assimilation of carbohydrates, and leaf photosynthesis. Similarly, the *ItfSS5* and *ItbSS5* expression was highest in *I. trifida* and *I. triloba* root1 (Figure 5B,C); it also had higher expression in leaves and stems; the expression levels of *SSs* in storage roots and leaves were different from those in sweet potato; and interestingly, the relative expression levels of flower buds were higher in *I. trifida* and *I. triloba*. It is also worth noting that both the *ItfSS5* and *ItbSS5* genes were highly expressed in both *I. trifida* and *I. triloba*.

Starch plays an important role in balancing growth and carbon assimilation processes, especially under stress conditions, and plants can mobilize starch reserves to release energy, sugars, and derived metabolites to mitigate the effects of stress [36]. It accumulated during the photoperiod and was degraded at night to support respiration and growth. In maize, starch synthesis helped plants maintain leaf growth and facilitate carbon uptake under drought conditions [35]. Similar conclusions were also made in faba bean [37], tomato [38], reed [39], wheat [40,41], rice [42], and sorghum [43]. In our study, *IbSS6* and *IbSS1* showed high expression under both salt and drought stress (Figure 13), while *ItfSS5*, *ItfSS8*, *ItbSS5,* and *ItbSS8* showed high expression under both salt and drought stress conditions (Figure 14). Thus, *SS* may play a crucial role in the salt and heat stress responses of *I. batatas*, *I. trifida,* and *I. triloba*.

In addition to salt and drought, cold stress and potassium deficiency also had important effects on sweet potato yield and quality. Using two varieties of cold-resistant “Liaohanshu 21” and cold-sensitive “Shenshu 28”, we found that *IbSS4* and *IbSS6* were upregulated in both varieties (Figure 9). High expression of *ItfSS5* and *ItbSS2* in two closely related species (Figure 10) suggests that SS may play an important role in the response of sweet potato and its relatives to cold stress in the evolutionary process. When two varieties of low-potassium-tolerant “Xushu 32” and low-potassium-sensitive “Ningzishu 1” were used, we found that all *IbSSs* genes were up-regulated, except for *IbSS4*, whose expression was down-regulated in both varieties (Figure 6). These results suggest that *SS* may have a positive effect on sweet potato under cold stress and potassium deficiency stress.

Finally, we examined the expression of different places in “P32” and “J26” (large storage roots, small storage roots, firewood root, and fibrous root) by qPCR. In large storage roots and small storage roots (root1 and root2), *IbSS6* in the GBSS group had a higher expression in “J26” exhibiting more amylose, according to the results of other studies, i.e., when GBSS expression was higher, the content of rectilinear starch was also higher [1]. *IbSSs* showed higher expressions in “P32” than “J26”. *IbSS3*, *IbSS4*, and *IbSS5* have higher expression in the stem and leaf (Figure 15), suggesting that these SS genes in sweet potato may affect carbohydrate transport, assimilation, and leaf photosynthesis. According to our cluster analysis and existing articles, it was known that *SS* genes not only play important roles in starch synthesis but also play key roles in abiotic stress response [44,45,46].

### 3.3. Involvement of SS in Hormonal Responses in Sweet Potato and Two Closely Related Species

Gene expression was usually regulated by cis-acting elements, and *SS* genes were involved in a number of physiological and biochemical processes such as plant growth and development, signal transduction, abiotic stress response, and hormone regulation [47]. A large number of cis-acting elements, including core promoter elements, light-responsive elements, hormone-responsive elements, stress-responsive elements, and unknown elements, have been identified in the promoter regions of *I. batatas*, *I. trifida*, and *I. triloba* (Figure 4). In this study, we found abundant ABA, GA, PHY, CAS, and MeJA response elements in the *IbSS*, *ItfSS*, and *ItbSS* promoters. Through expression analysis, we found that *IbSSs* showed different expression patterns. Among them, *IbSS2*, *IbSS4*, *IbSS6,* and *IbSS7* were all up-regulated under ABA, SA, and MeJA treatments (Figure 7). After ABA, GA, and IAA treatments, *ItfSS1*, *ItfSS5*, *ItfSS7*, *ItfSS8*, *ItbSS1*, *ItbSS2*, *ItbSS3*, *ItbSS4*, *ItbSS6*, *ItbSS7*, and *ItbSS8* show different degrees of up-regulation (Figure 8A,B). These results suggest that *SS* may be involved in hormone crosstalk in sweet potato and its diploid wild relatives.

Differences in expression patterns of *SS* genes in sweet potato and its two diploid relatives might provide potential candidate genes for further functional characterization and provide a certain reference significance for the molecular breeding of sweet potato in the future.

## 4. Materials and Methods

### 4.1. Identification of SS Genes Members in Sweet Potato

The protein sequences and GFF gene annotation files of *I. batatas*, *I. trifida*, and *I. triloba* were obtained from the Sweet Potato Genomics Resource (Sweetpotato Genomics Resource (uga.edu), accessed on 17 July 2023) and Ipomoea Genome Hub (https://ipomoea-genome.org/, accessed on 17 July 2023). The Cryptomarkov model (HMM) profile of the glycosyltransferase domain (PF08323 and PF00534) was downloaded from the Pfam (Pfam: Home page (xfam.org)) database and used to investigate all protein sequences in *I. batatas*, *I. trifida,* and *I. triloba*. The *Arabidopsis*, *S. bicolor*, and *S. lycopersicum* SS protein sequences were downloaded at Ensembl Plants (https://plants.ensembl.org/index.html, accessed 20 September 2023). The *O. sativa* SS protein sequences were obtained from the National Rice Data Center (RiceData = Rice Gene Database, accessed on 22 June 2023). The HMMER Search program in Tbtools software was used to preliminarily screen the *SS* candidate genes. All putative *SS* were further screened using SMART (SMART: Main page (embl.de), accessed on 22 June 2023) and finally further validated by the conserved database CDD (Welcome to NCBI Batch CD-search (nih.gov), accessed on 22 June 2023) [48]. Finally, with the full-length transcriptome of third-generation sequenced “Guangshu 87” as a reference, the candidate sweet potato SS family members were manually corrected using SnapGene software 6.0.2. Screened genes of *IbSS*, *ItfSS,* and *ItbSS* were named sequentially based on chromosomal location. The physicochemical properties of *IbSS*, *ItfSS,* and *ItbSS* proteins were analyzed based on the basic information of the protein sequences determined by our laboratory in the company and the Expasy online tool (SIB Swiss Institute of Bioinformatics Expasy, accessed on 22 June 2023) [49]. TBtools software Version 2.0 was used for visualization and mapping [50].

### 4.2. Phylogenetic Analysis of Sweet Potato SS Genes

Using ClustalW Muscle default parameters, we performed multiple sequence comparisons of *I. batatas*, *I. trifida*, *I. triloba*, *A. thaliana*, *S. bicolor*, *O. sativa*, and *S. lycopersicum,* constructed a phylogenetic tree using MEGA7[51], and grassed on the classification of SS in *A. thaliana*, *S. bicolor*, *O. sativa*, and *S. lycopersicum* to classify the sweet potato, triticale ragweed, and triticale yam *SS* genes. Finally, the evolutionary trees were landscaped using the EvoView online tool (EvolView: my projects and trees (evolgenius.info), accessed on 22 June 2023).

### 4.3. Conserved Motifs and Exon–Intron Structure Analysis of Sweet Potato SS

Using the MEME online tool (MEME—Submission form (meme-suite.org), accessed on 22 August 2023) analyzed conserved motifs of SS proteins, and the maximum number of motifs parameter set to 10. The required protein sequences were put into CDD (NCBI Conserved Domain Search (nih.gov), accessed on 15 October 2023) to obtain the corresponding files. Finally, TBtools software Version 2.0 was used to visualize and analyze the Motif predictions of *IbSS*, *ItfSS*, and *ItbSS*, as well as the NCBI results.

### 4.4. Analysis of the Promoter Cis-Acting Element of the Sweet Potato SS Genes

Based on the genome annotation information and whole genome sequences of *I. batatas*, *I. trifida,* and *I. triloba*, sequences 2000 bp upstream of the CDS of *I. batatas*, *I. trifida*, and *I. triloba* were extracted as promoter regions, and the promoter sequences were submitted to the PlantCare website (PlantCARE, a database of plant promoters and their cis-acting regulatory elements (ugent.be), accessed on 19 October 2023) for cis-acting element prediction. The PlantCare results were simplified, and the cis-acting elements of the SS promoter region were obtained using TBtools software Version 2.0.

### 4.5. Analysis of SS Genes Expression in Sweet Potato

For the analysis of sweet potato *SSs* expression profiles, a total of four transcriptome bio project datasets were selected. The data included three bio project datasets obtained from the NCBI database: PRJNA1013090 focusing on low potassium, PRJNA511028 on hormone, and PRJNA987163 on cold. Additionally, an in-house (unpublished) dataset was included, specifically addressing sweet potato heat treatment. Within these datasets, different sweet potato varieties were utilized for specific treatments. For hormonal treatment, “Xushu 18” was chosen, while cold-tolerant “Liaohanshu 21” and cold-sensitive “Shenshu 28” were used for cold treatment. Low-tolerant “Xushu 32” and low-k-sensitive “Ningzishu 1” were selected for low-potassium treatment, and heat-tolerant “Guangshu 87” and heat-sensitive “Ziluolan” were employed for heat treatment. Moreover, the gene expression data for *I. trifida* and *I. triloba* were obtained from the Sweet Potato Genomics Resource (http://sweetpotato.plantbiology.msu.edu/, accessed on 17 July 2023). The expression maps were created with TBtools software version 2.0.

### 4.6. qRT-PCR Detection of SS Genes in Sweet Potato

Eight different parts of two varieties, high amylose “J26” and low amylose “P32”, were used to analyze the expression levels. Total RNA was extracted using the TRIzol method (Invitrogen, Carlsbad, CA, USA). The qRT-PCR experiments were an improvement over previous studies [52]. The experiments were performed by fluorescence quantification kits, with 3 replicates per sample. For qRT-PCR analysis, the 20 µL total reaction quantity of each sample contained 2 µL cDNA template, 2 µL (10 µmol L^−1^) forward and reverse gene-specifc primers, 10 µL 2 × SYBR Green qPCR mix, and 4 µL sterile ddH_2_O. The qRT-PCR reaction was conducted using the Bio-Rad system with the following thermal cycle conditions: 3 min of pre-degeneration at 95 °C, followed by 40 cycles of denaturation at 95 °C for 10 s, and annealing at 60 °C for 30 s. The reaction was completed with a 5-s step at 65 °C and a cooling rate of 0.5 °C to reach 95 °C. The results were averaged for gene expression analysis. The relative expression of *SSs* in sweet potato was calculated by the 2^−ΔΔCt^ method. Primers used in the study are listed in Appendix A, where the *IbARF* gene is an internal reference [53].

### 4.7. Protein–Protein Interaction Analysis of SS Protein in Sweet Potato

Using the default parameters, the online STRING database (STRING: functional protein association networks (string-db.org), accessed on 22 July 2023) was used to predict and execute protein–protein interaction networks based on the *Arabidopsis* homolog of the sweet potato *SS* protein. Finally, Cytoscape software was used for visualization and mapping.

## 5. Conclusions

Eight *SSs* were identified in *I. batatas*, *I. trifida*, and *I. triloba*, respectively. Their protein physicochemical properties, chromosomal localization, phylogenetic relationship, gene structure, promoter cis-elements, coercion expression, and protein interaction network and expression patterns were systematically analyzed. *SS* genes may be involved in various hormone and abiotic stress responses to regulate sweet potato growth and development. For example, *IbSSs* were highly expressed in sweet potato tuber root, suggesting that *IbSSs* may play an important role in sweet potato yield and quality. The *IbSS6* gene was highly expressed in the face of various stress treatments; other genes are also expressed to varying degrees; among them, *IbSS1* and *IbSS8* are upward in the face of coercion. These candidate genes merit further investigation. The findings in this study will help us better understand the biological functions of *SSs* in coping with climate change and identify possible candidate genes for enhancing field and abiotic stress tolerance in sweet potato and its two diploid relatives.

## Figures and Tables

**Figure 1 genes-15-00400-f001:**
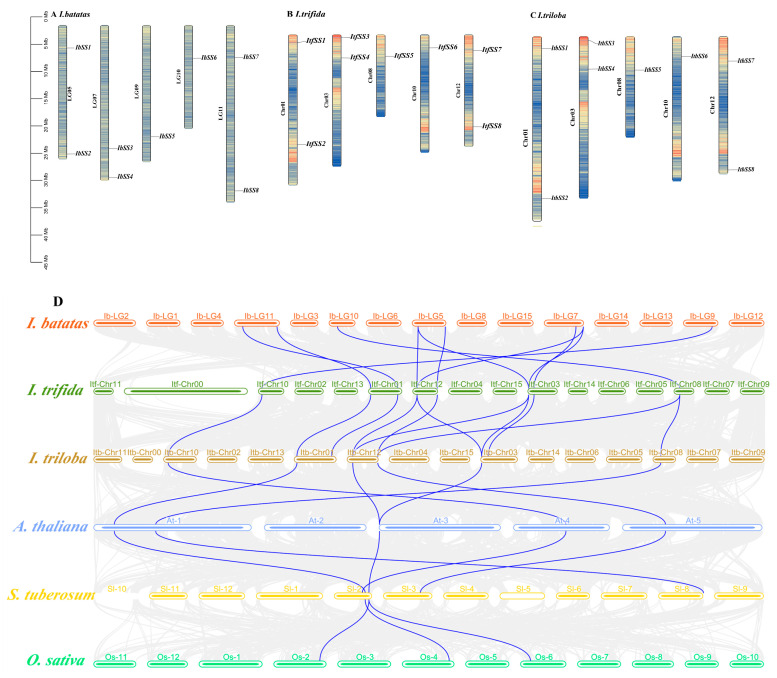
Chromosome localization and distribution of *SSs* in *I. batatas*, *I. trifida,* and *I. triloba* (**A**–**C**). Chromosome numbers are shown on the left and gene names on the right. (**D**) Collinearity analyses of *SSs* in *I. batatas*, *I. trifida*, *I. triloba*, *A. thaliana*, *S. lycopersicum*, and *O. sativa* are indicated by different colors. Orange, light green, brown, light blue, yellow, and green, respectively, represent *I. batatas*, *I. trifida*, *I. triloba*, *A. thaliana*, *S. lycopersicum*, and *O. sativa.*

**Figure 2 genes-15-00400-f002:**
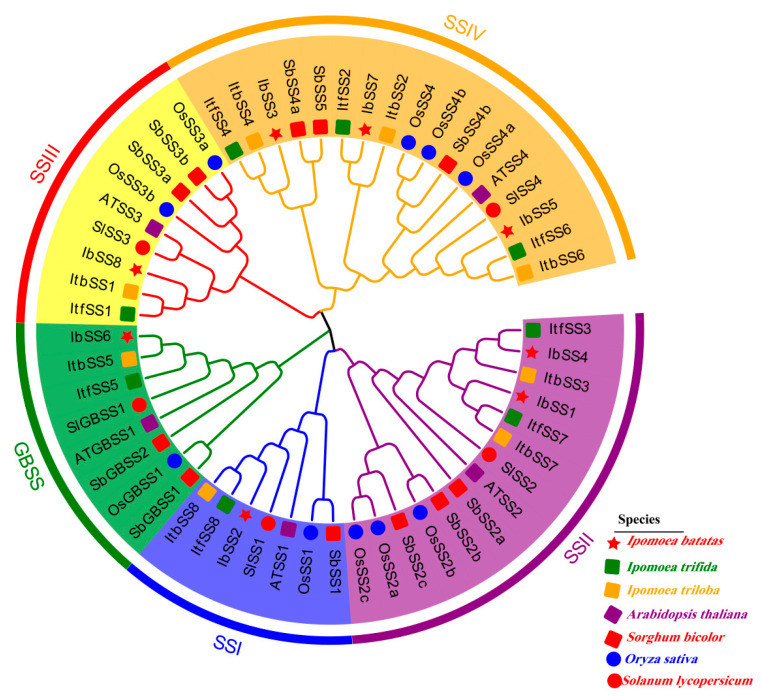
Phylogenetic analyses of *SSs* in *I. batatas*, *I. trifida*, *I. triloba*, *A. thaliana*, *S. bicolor*, *O. sativa*, and *S. lycopersicum*. Based on the evolutionary analysis, the 24 *SSs* were categorized into five groups (I, II, III, and IV are indicated in blue, purple, red, and orange, respectively), with green indicating granule-bound starch synthases. Red pentagrams indicate eight *IbSS* in *I. batatas*, green squares indicate eight *ItfSS* in *I. trifida*, orange squares indicate eight *ItbSS* in *I. triloba*, purple squares indicate five *ATSS* in *A. thaliana*, red squares indicate 11 *SbSS* in *S. bicolor*, blue circles indicate 10 *OSSs* in *O. sativa*, and red squares indicate five *SlSS* in *S. lycopersicum.*

**Figure 3 genes-15-00400-f003:**
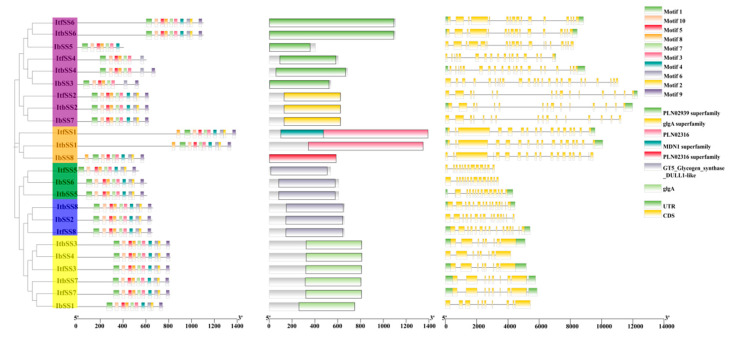
Analysis of conserved motifs and exon–intron structure of starch synthase (*SS*) in three species, *I. batatas*, *I. trifida*, and *I. triloba*, revealed interesting findings. The phylogenetic tree indicated that *SS* can be classified into seven distinct subgroups, with the top ten most prevalent subgroups identified. The *SS* were divided into seven subgroups and the ten conserved motifs are shown in different colors, purple: SSIV, orange: SSIII, green: GBSS blue: SSI yellow: SSII. The far right represents the exon–intron structure of *SS*.

**Figure 4 genes-15-00400-f004:**
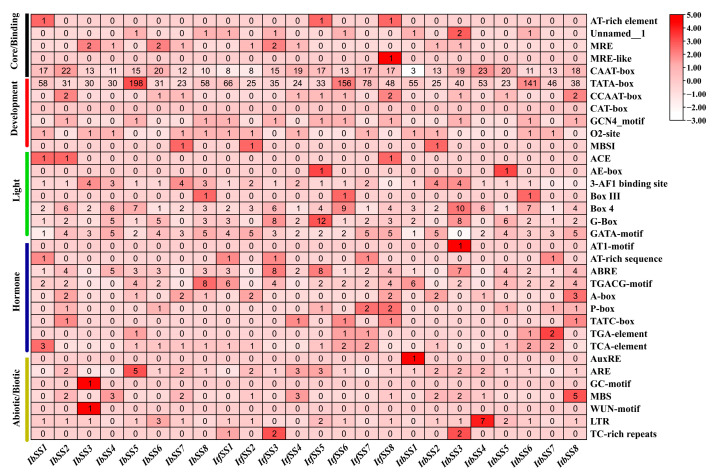
Analysis of cis−elements in the promoters of *SS* from *I. batatas*, *I. trifida*, and *I. triloba*. The cis−elements were categorized into six groups. The intensity of the red hue indicates the quantity of cis-elements in the *SS* promoters.

**Figure 5 genes-15-00400-f005:**
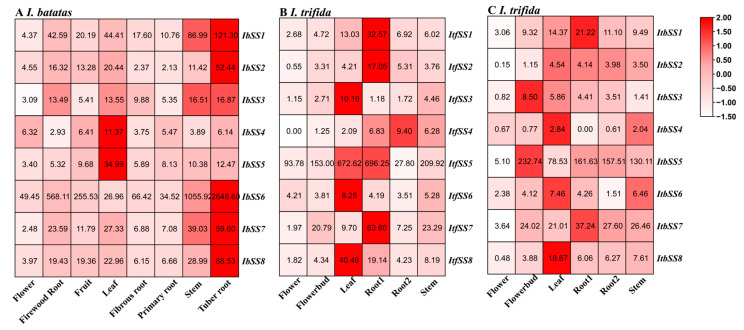
Expression analysis of *SS* in different tissues of *I. batatas*, *I. trifida*, and *I. triloba* using RNA-seq. (**A**) Expression analysis of *IbSSs* in different tissues. (**B**) Expression patterns of *ItfSSs* in the flower, flowerbud, leaf, stem, root1, root2, and stem of *I. trifida*. (**C**) Expression patterns of *ItbSSs* in the flower, flowerbud, leaf, stem, root1, root2, and stem. The FPKM values were shown in the boxes.

**Figure 6 genes-15-00400-f006:**
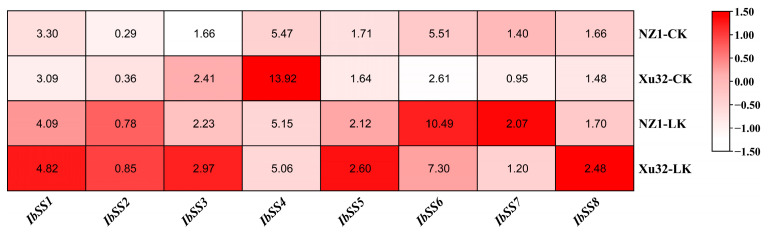
Expression analysis of sweet potato *SS* genes under potassium deficiency as determined via RNA-seq. NZ1: “Ningzishu 1”; Xu32: “Xushu 32”. The FPKM values are shown in the color blocks.

**Figure 7 genes-15-00400-f007:**
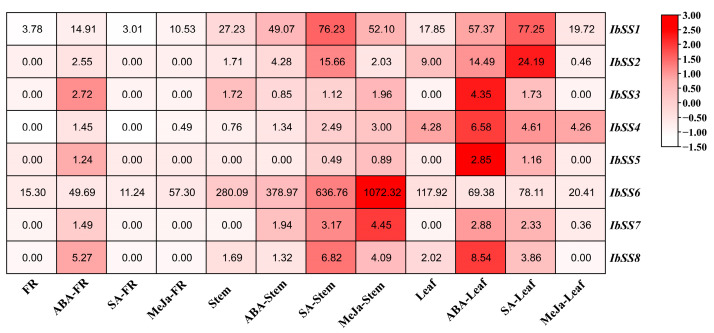
Analysis of the expression of genes related to sweet potato *SSs* in fibrous roots (FR), stems, and leaves of sweet potato after hormone treatments using RNA-seq technology. The FPKM values are displayed in colored blocks.

**Figure 8 genes-15-00400-f008:**
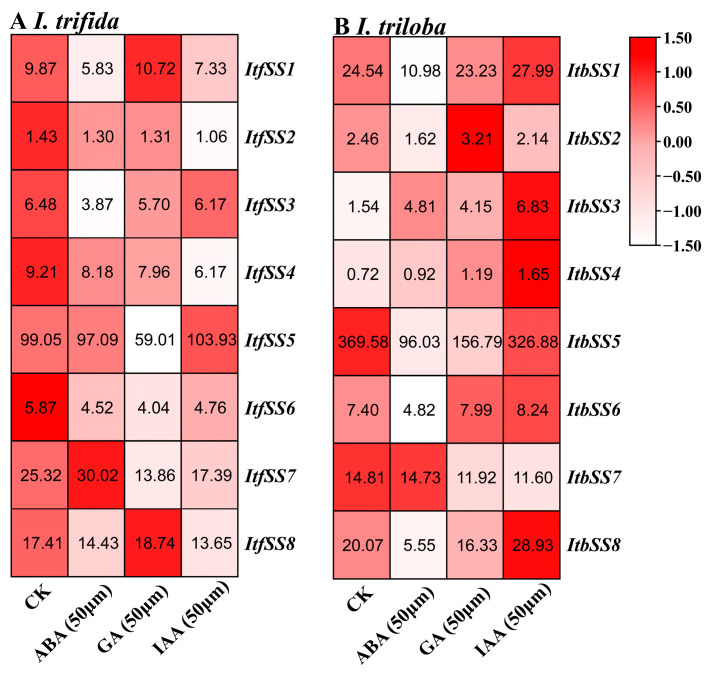
Expression analysis of *ItfSSs* (**A**) and *ItbSSs* (**B**) in response to different hormones ABA (50 μm), GA (50 μm) and IAA (50 μm) as determined by RNA-seq. FPKM values are shown in the boxes.

**Figure 9 genes-15-00400-f009:**
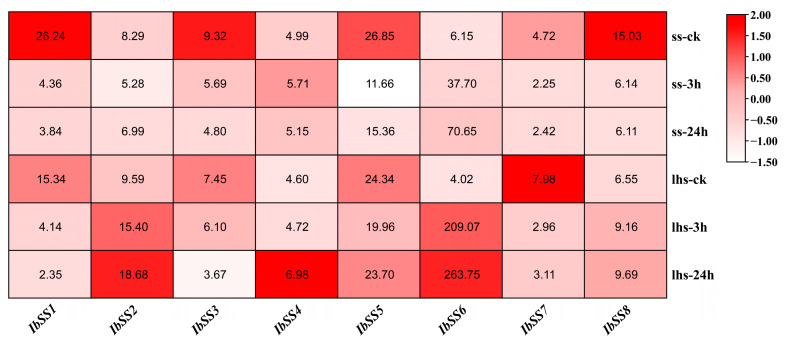
RNA-seq analysis revealed the gene expression profiles of sweet potato *SS* genes under cold stress. The cold-sensitive variety “Shenshu 28” was compared to the cold-tolerant variety “Liaohanshu 21”. The color block displays the FPKM values for each sample.

**Figure 10 genes-15-00400-f010:**
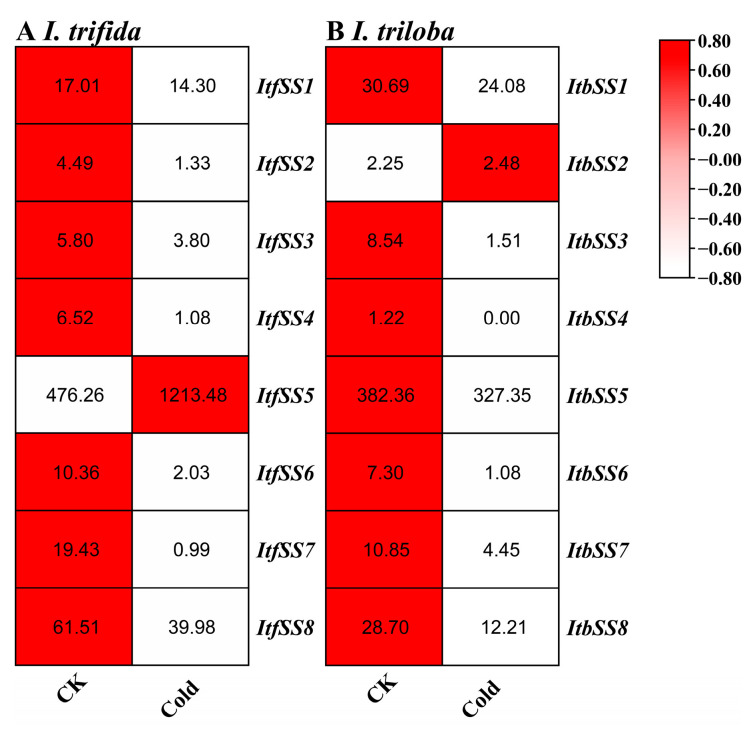
Expression analysis of *ItfSSs* (**A**) and *ItbSSs* (**B**) in response to cold stress as determined by RNA-seq. FPKM values are shown in the boxes.

**Figure 11 genes-15-00400-f011:**
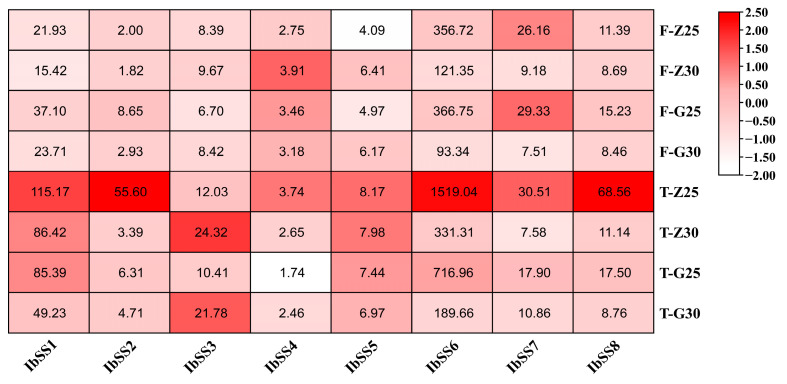
Gene expression patterns of sweet potato *SS* genes under heat stress were determined using RNA-seq analysis. F represents fibrous roots, T represents tuberous roots, Z represents the heat-sensitive “Ziluolan”, and G represents the heat-tolerant “Guangshu 87”. The heatmap displays the FPKM values for each gene.

**Figure 12 genes-15-00400-f012:**
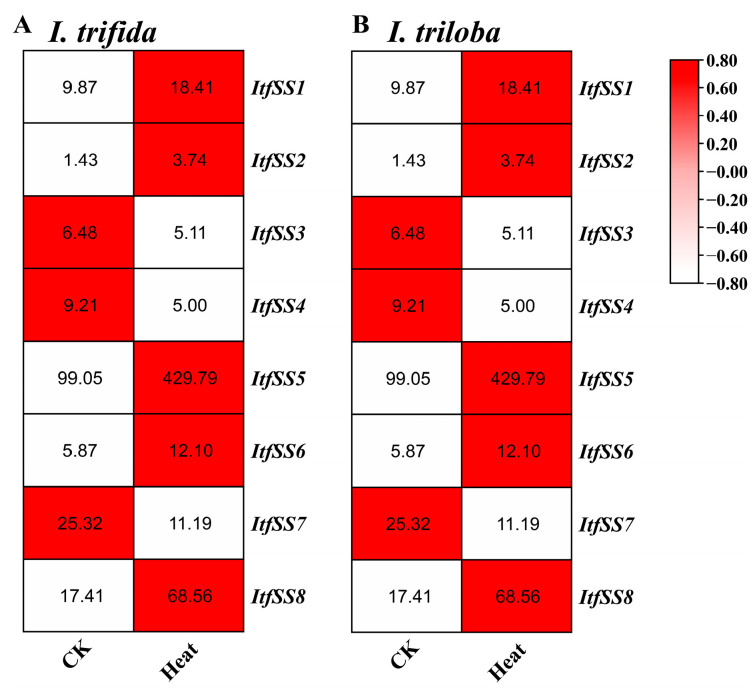
Expression analysis of *ItfSSs* (**A**) and *ItbSSs* (**B**) in response to heat stress as determined by RNA-seq. FPKM values are shown in the boxes.

**Figure 13 genes-15-00400-f013:**
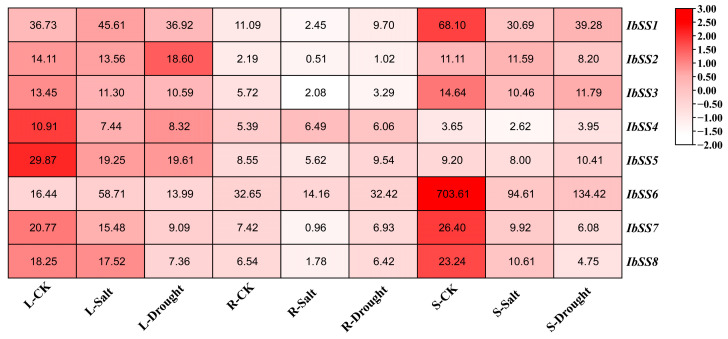
Expression patterns of sweet potato *SS* genes under salt and drought stress as determined via RNA-seq. L: leaves; R: primary roots; S: stems. The FPKM values are shown in the color blocks.

**Figure 14 genes-15-00400-f014:**
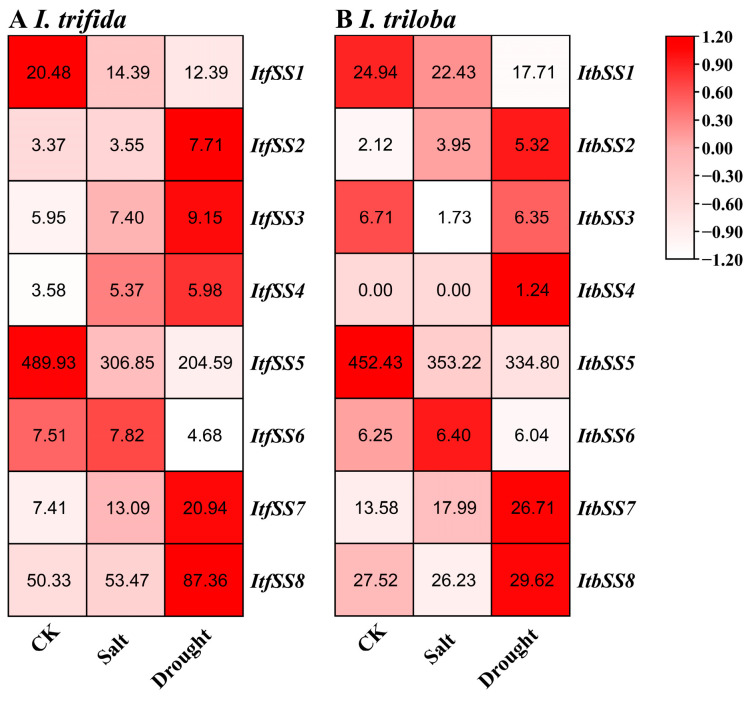
Expression analysis of *ItfSSs* (**A**) and *ItbSSs* (**B**) in response to salt and drought stress as determined by RNA-seq. FPKM values are shown in the boxes.

**Figure 15 genes-15-00400-f015:**
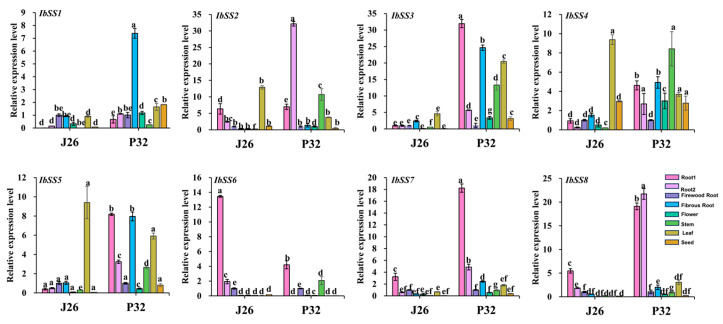
Expression analysis of *SS* genes in sweet potato different tissue sites. J26: “Jishu 26”; P32: “Pushu 32”; root1: large tuber roots; root2: small tuber roots. Lowercase letters indicate a significant difference in each *IbSS* at *p* < 0.05 as determined by one-way ANOVA, with Tukey’s post hoc test conducted afterwards.

**Figure 16 genes-15-00400-f016:**
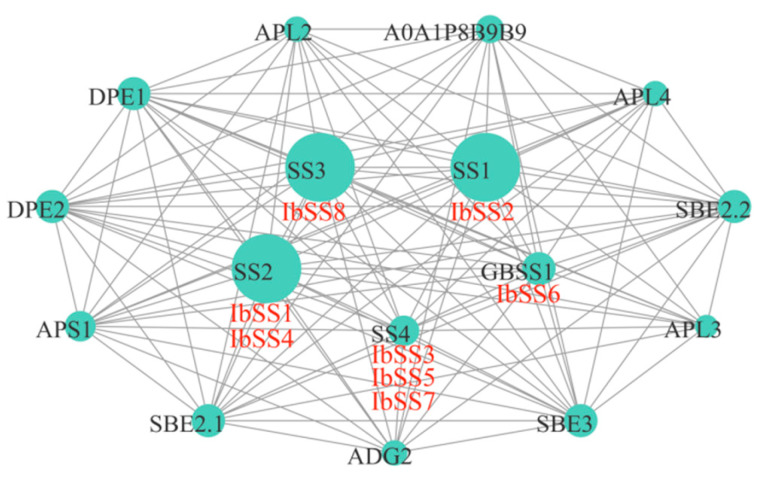
Mapping the sweet potato *SS* protein–protein interaction network based on homolog proteins in *Arabidopsis*. The size of the circle and the thickness of the line represent the degree of interaction.

**Table 1 genes-15-00400-t001:** Characterization of *SS* in sweet potato, *I. trifida*, and *I. triloba.*

Accession Number	Gene Name	Chromosome	Location		Mw(kDa)	Pl	Hydropathicity	Amion Acid	Instability
			Start	End					
AAC19119.1	*IbSS1*	LG5	5,188,689	5,194,313	85.28	5.31	−0.388	770	44.68
OR861658	*IbSS2*	LG5	29,389,428	29,394,162	71.72	5.17	−0.147	651	33.61
OR861659	*IbSS3*	LG7	28,141,052	28,152,901	107.98	6.66	−0.109	953	41.17
OR861660	*IbSS4*	LG7	34,676,444	34,680,867	89.60	5.67	−0.367	811	40.82
OR861661	*IbSS5*	LG9	25,462,397	25,471,028	116.65	5.18	−0.023	1037	45.73
BAI83439.1	*IbSS6*	LG10	7,535,337	7,539,015	70.69	7.15	−0.023	641	30.32
OR861662	*IbSS7*	LG11	7,369,578	7,381,372	70.83	5.88	−0.112	633	35.68
OR861663	*IbSS8*	LG11	37,816,551	37,826,370	157.43	5.11	−0.561	1402	42.03
OR861664	*ItfSS1*	LG1	2,007,039	2,016,615	156.75	4.96	−0.571	1391	41.2
OR861665	*ItfSS2*	LG1	27,774,044	27,786,340	70.40	6.06	−0.143	627	35.3
OR861666	*ItfSS3*	LG3	728,515	733,683	89.53	5.67	−0.374	811	40.48
OR861667	*ItfSS4*	LG3	5,894,206	5,901,262	68.06	6.56	−0.197	604	39.81
OR861668	*ItfSS5*	LG8	5,505,555	5,508,685	59.38	6.34	−0.035	538	25.11
OR861669	*ItfSS6*	LG10	3,353,952	3,362,791	124.61	5.17	−0.467	1104	46.85
GLL26599.1	*ItfSS7*	LG12	4,022,872	4,028,737	89.40	5.51	−0.377	811	42.61
OR861670	*ItfSS8*	LG12	4,028,737	23,299,232	72.82	5.28	−0.248	657	31.85
XP_031110077.1	*ItbSS1*	LG1	2,448,185	2,458,262	152.42	5.13	−0.579	1349	41.74
OR861671	*ItbSS2*	LG1	33,298,263	33,310,246	70.57	6.16	−0.129	627	35.56
XP_031107980.1	*ItbSS3*	LG3	833,120	838,217	89.69	5.64	−0.377	813	42.68
XP_031109143.1	*ItbSS4*	LG3	6,699,089	6,708,023	77.55	6.06	−0.233	685	42.36
XP_031124794.1	*ItbSS5*	LG8	6,898,989	6,903,301	66.70	8.31	−0.075	608	26.11
XP_031130996.1	*ItbSS6*	LG10	4093,699	4,102,135	124.59	5.17	−0.465	1104	46.85
XP_031094614.1	*ItbSS7*	LG12	5,048,708	5,054,476	88.81	5.50	−0.335	806	41.47
XP_031096470.1	*ItbSS8*	LG12	27,425,473	27,429,928	73.12	5.39	−0.239	660	32.56

Note: Genes such as *IbSS1*, *IbSS6*, *ItfSS7*, *ItbSS1*, *ItbSS3*, *ItbSS4*, *ItbSS5*, *ItbSS6*, *ItbSS7*, and *ItbSS8* were uploaded in National Center for Biotechnology Information (NCBI), PI: isoelectric point.

## Data Availability

Data are contained within the article or Appendix A.

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
