# Peer review of "Genome-Wide Identification and Expression Analysis of the Starch Synthase Gene Family in Sweet Potato and Two of Its Closely Related Species"

_genes, 2024, doi:10.3390/genes15040400_

Round 1

Reviewer 1 Report

Comments and Suggestions for Authors

The reported research provides useful information on starch synthase gene in sweetpotato.

However, the manuscript needs some revision in terms of clarity and accuracy. 

Specifically,  the literature review is not complete.  The authors mentioned that “Although the SS family has been studied in many crops, it has not yet been explored in sweet potato” (Line 10-11).

This statement is not accurate.  Numerous studies have been reported on sweetpotato starch synthase.  The authors need to properly review the published literature, and provide a comprehensive summary.   

In addition,  the English writing and accuracy need to be improved.  In the Introduction, the section  about sweet potato and sweetpotato starch needs further polishment. 

I will cite two examples here:

 Line 64-65,  “..... which was grown on a large scale in China because of its high yield and resistance”.   Please replace “in China” with “in the world” , and replace “resistance” with “resistance to abiotic stresses”. 

Line 66,  “It is an important root crop, and the main component of its tuberous roots….”.   Please replace “tuberous roots” with  “storage roots”.   Sweetpotato is a root crop, not a tuber crop.

………

Comments on the Quality of English Language

In addition,  the English writing and accuracy need to be improved.  In the Introduction, the section  about sweet potato and sweetpotato starch needs further polishment. 

I will cite two examples here:

 Line 64-65,  “..... which was grown on a large scale in China because of its high yield and resistance”.   Please replace “in China” with “in the world” , and replace “resistance” with “resistance to abiotic stresses”. 

Line 66,  “It is an important root crop, and the main component of its tuberous roots….”.   Please replace “tuberous roots” with  “storage roots”.   Sweetpotato is a root crop, not a tuber crop.

………

Author Response

Response to Reviewer 1 Comment

1. Summary

Thank you very much for taking the time to review this manuscript in your busy schedule. All of authors have carefully read the comments that you have given us, and have discussed and revised each of these issues. The following is my list of revisions. In addition, we have resubmitted a new manuscript in the revised state, with the revisions highlighted in red. If there are any incorrect answers or questions in the manuscript, please do not hesitate to let us know.

2. Point-by-point response to Comments and Suggestions for Authors

Comments 1: Line 10-11: “Although the SS family has been studied in many crops, it has not yet been explored in sweet potato” 

Response 1: Thank you for pointing this out. We edited “Although the SS family has been studied in many crops, it has not yet been explored in sweet potato” to “Although the SS family has been studied in many crops, it has not been fully identified in sweet potato and its two related species” (Lines 10-11 in the revised manuscript).

Comments 2: Lin64-65: “..... which was grown on a large scale in China because of its high yield and resistance”. Please replace “in China” with “in the world”, and replace “resistance” with “resistance to abiotic stresses”. 

Response 2: Thanks for your suggestions. We replaced “in China” with “in the world”(Lines 67 in the revised manuscript), and replace “resistance” with “resistance to abiotic stresses” (Lines 68 in the revised manuscript). 

Comments 3: Lin66: “It is an important root crop, and the main component of its tuberous roots….”.   Please replace “tuberous roots” with “storage roots”.   Sweet potato is a root crop, not a tuber crop.

Response 3: Thanks for your reminder. “tuberous roots” was changed into “storage roots”(Lines 68 in the revised manuscript)

 3. Response to Comments on the English writing and accuracy need to be improved.

Point 1: many sentences required corrections

Response 1: Thanks for your suggestions. We tried our best to improve the manuscript and made some changes. These changes will not influence the content and framework of the paper. We did not list the changes but marked them in red in the revised manuscript. We appreciate your warm work earnestly and hope that the correction will meet with approval.

Sincerely,

Zongjian Sun

Reviewer 2 Report

Comments and Suggestions for Authors

The objective of this study was to characterize both the sequences and expression patterns of diverse genes implicated in starch synthesis. The conducted phylogenetic analyses are innovative as they offer insights into the genetic similarities among the sequences of various sucrose synthases (GBSS, SSI, SSII, SSIII, and SSIV) across the three sweet potato species (I. batatas, I. trifida, and I. triloba), juxtaposed with those of other significant species such as A. thaliana, O. sativa, and L. lycopersicum. The multiple expression profiles presented, derived from RNA-seq data obtained from various transcriptomic projects, delineated distinctive expression patterns across different plant organs and in response to diverse environmental stressors such as potassium deficiency, salinity, cold stress, etc. Moreover, the authors conducted an RT-PCR study, analyzing variations in the expression of various starch synthase genes within the organs of two contrasted varieties. These results hold great promise, revealing promising avenues for further research, necessitating validation of the proposed hypotheses across a larger cohort of contrasting genotypes.

Specific comments:
In Figure 1 part D, it is advised to relocate the red arrows, which are positioned on the chromosomes, making the chromosome names illegible.
In line 30, "amylopectin (AM)" should be changed to "amylose (AM) ".
In the Materials and Methods section, paragraph 4.6 (line 499), it is essential to add a better description of experimental conditions, including details on the PCR program, reaction conditions, and probe sequences. This information is crucial to ensure result reproducibility and methodological transparency. In the absence of this, it would be appropriate to reference a publication where the various experimental conditions are exhaustively detailed.

Author Response

Response to Reviewer 2 Comments

1. Summary

Thank you very much for taking the time to review this manuscript in your busy schedule. All of authors have carefully read the comments that you have given us, and have discussed and revised each of these issues. The following is my list of revisions. In addition, we have resubmitted a new manuscript in the revised state, with the revisions highlighted in red. If there are any incorrect answers or questions in the manuscript, please do not hesitate to let us know.

2. Point-by-point response to Comments and Suggestions for Authors

Comments 1: Line 119-120: “In Figure 1 part D, it is advised to relocate the red arrows, which are positioned on the chromosomes, making the chromosome names illegible.” 

Response 1: Thank you for pointing this out. We've removed the "red arrows". (Lines 119-120 in the revised manuscript).

Comments 2: Line 30: “amylopectin (AM)" should be changed to "amylose (AM)”. 

Response 2: Thanks for your suggestions. We replaced “amylose (AM)” with “amylopectin (AM)” (Lines 30 in the revised manuscript) 

Comments 3: Line 499: “in the Materials and Methods section, paragraph 4.6 (line 499), it is essential to add a better description of experimental conditions, including details on the PCR program, reaction conditions, and probe sequences”.

Response 3: Thanks for your reminder. We've added relevant experimental conditions, including details of the PCR program, reaction conditions, and probe sequences (Lines 520-533 in the revised manuscript, among them, the probe sequence is in the supplementary file).

Sincerely,

Zongjian Sun
